# Akaike’s Bayesian Information Criterion for the Joint Inversion of Terrestrial Water Storage Using GPS Vertical Displacements, GRACE and GLDAS in Southwest China

**DOI:** 10.3390/e21070664

**Published:** 2019-07-07

**Authors:** Yongxin Liu, Hok Sum Fok, Robert Tenzer, Qiang Chen, Xiuwan Chen

**Affiliations:** 1School of Earth and Space Sciences, Peking University, Beijing 100871, China; 2School of Geodesy and Geomatics, Wuhan University, Wuhan 430079, China; 3Department of Land Surveying and Geo-informatics, The Hong Kong Polytechnic University, Kowloon, Hong Kong, China; 4Geophysics Laboratory, Faculty of Science, Technology and Communication, University of Luxembourg, 2, avenue de l’Université, L-4365 Esch-sur-Alzette, Luxembourg; 5Key Laboratory of Geospace Environment and Geodesy, Ministry of Education, Wuhan University, Wuhan 430079, China; 6Engineering Research Center of Earth Observation and Navigation (CEON), Ministry of Education of the PRC, No. 5 Yiheyuan Road, Haidian District, Beijing 100871, China

**Keywords:** terrestrial water storage inversion, GPS, GRACE, GLDAS, Akaike’s Bayesian information criterion

## Abstract

Global navigation satellite systems (GNSS) techniques, such as GPS, can be used to accurately record vertical crustal movements induced by seasonal terrestrial water storage (TWS) variations. Conversely, the TWS data could be inverted from GPS-observed vertical displacement based on the well-known elastic loading theory through the Tikhonov regularization (TR) or the Helmert variance component estimation (HVCE). To complement a potential non-uniform spatial distribution of GPS sites and to improve the quality of inversion procedure, herein we proposed in this study a novel approach for the TWS inversion by jointly supplementing GPS vertical crustal displacements with minimum usage of external TWS-derived displacements serving as pseudo GPS sites, such as from satellite gravimetry (e.g., Gravity Recovery and Climate Experiment, GRACE) or from hydrological models (e.g., Global Land Data Assimilation System, GLDAS), to constrain the inversion. In addition, Akaike’s Bayesian Information Criterion (ABIC) was employed during the inversion, while comparing with TR and HVCE to demonstrate the feasibility of our approach. Despite the deterioration of the model fitness, our results revealed that the introduction of GRACE or GLDAS data as constraints during the joint inversion effectively reduced the uncertainty and bias by 42% and 41% on average, respectively, with significant improvements in the spatial boundary of our study area. In general, the ABIC with GRACE or GLDAS data constraints displayed an optimal performance in terms of model fitness and inversion performance, compared to those of other GPS-inferred TWS methodologies reported in published studies.

## 1. Introduction

Water resources are indispensable in agriculture and for other sustainable socio-economic activities. Unfortunately, water scarcity is seemingly a near future possibility [1]. This problem is likely to delay and threaten future developments in agriculture [2]. Therefore, it is essential that we comprehend the four key components of the hydrologic cycle (i.e., precipitation, evapotranspiration, runoff, and terrestrial water storage (TWS)), among which TWS is an essential part of the global hydrologic cycle and the Earth’s climate system [3]. 

Currently, there are two main approaches to acquiring TWS data: (1) satellite gravimetry, such as the Gravity Recovery and Climate Experiment (GRACE) [4] and (2) data assimilation and modeling techniques, such as the Global Land Data Assimilation System (GLDAS) [5]. These sources have provided continuous information about TWS variations with global coverage. However, GRACE has coarse spatial resolution (i.e., larger than 100,000 km^2^) and low temporal resolution (i.e., monthly) that limit its applications in large regions and long-periodic phenomena. In addition, various types of errors affect the accuracy of GRACE observations, such as spatial and temporal aliasing errors [6,7], leakage effects, and uncertainties in glacial isostatic adjustment correction [8]. Higher spatial (i.e., 0.25° × 0.25° grid) and temporal (i.e., 3 h) resolutions of TWS inferred from land assimilation systems, such as GLDAS, merely considers TWS as encompassing soil moisture within a depth of 2 m, snow, and canopy water, where groundwater, reservoirs and lakes are largely ignored.

In essence, GPS can be considered as a viable method to infer TWS data, when a relatively dense observation network is available. GPS can be used to accurately record the seasonal variations of vertical crustal displacements induced by TWS [9,10]. Several researchers have reported in recent literature that GPS vertical displacements (VD) highly correlate with the TWS data inferred from GRACE [11,12,13,14]. These studies revealed a potential to retrieve TWS using GPS. Given the well-known relationship of a crustal response to surface loading (e.g., TWS) formulated by the elastic loading theory [15], Argus et al. [16] first developed the inversion methodology of TWS using the relatively dense GPS network in California affiliated to Plate Boundary Observatory (PBO). Fu et al. [17] and Jin and Zhang [18] applied the above method in Washington and Oregon states, and the entire Southwestern United States, and compared the result with GRACE-inferred TWS. This method also provides a new perspective on achieving a relatively high spatio-temporal resolution of TWS depending on the spatial density and distribution of the continuous GPS network [19]. In addition, GPS is a ground-based observation technology that should be more sensitive to variations in the overall regional hydrological compartments, including groundwater, nearby reservoirs [20], snow [21], and runoffs [22]. This implies that a ground-based GPS-inferred TWS should reflect the regional characteristics of TWS distribution more realistically than inferred from space-borne GRACE [23].

Sparse and uneven distribution of GPS stations will lead to unstable inversion results. Thus, an appropriate inversion method and constraint strategies are essential for this purpose. Argus et al. [16] and Jin et al. [18] employed spatial smoothness as constraint equation via the Tikhonov regularization (TR) to reach an optimal solution by minimizing least-squares. However, GPS-inferred TWS yielded results much larger than those inferred from GRACE and GLDAS. Zhang et al. [24] reached optimal solutions by applying the Helmert variance component estimation (HVCE) as well as the minimum mean square error (MMSE) as a criterion. Despite a better performance achieved by HVCE, the solution is sensitive to an initial value that might cause its divergence. For the weighting among spatial-temporal smoothness and various observations, the Akaike’s Bayesian Information Criterion (ABIC), an objective method based on an entropy maximization principle, has been commonly applied for joint inversion of geodetic and seismic data and prior constraint in the finite fault slip distribution inversion and the rupture processes [25,26,27].

In this study, we propose the use of the ABIC technique for the TWS inversion of GPS VD with the minimum usage of GRACE or GLDAS data as “pseudo GPS sites” for a-priori information due to potentially sparse and uneven distribution of GPS stations within the region. This novel approach is tested using 34 continuous GPS stations located in southwest China where hydro-climatic characteristics manifest apparent seasonal signals (see Figure 8 from [28]). To verify the numerical performance of the proposed inversion technique and results, the proposed ABIC methodology is compared with the HVCE and TR techniques in terms of model fitness, inversion stability, and mutual comparison of GPS-inferred TWS with other GRACE-inferred and GLDAS-inferred TWS.

## 2. Study Area and Datasets

### 2.1. Geographic Background of Southwest China

The study area situated in southwest China comprises the Yunnan-Guizhou Plateau, with an altitude varying roughly from 1500 to 4000 m. The northeastern part of the study area covers the southeast Tibet Plateau, known as the Hengduan Mountains with a number of mountain glaciers [29]. This region is climate-driven and affected by seasonal monsoons [30]. Three large streams flow through this area, namely the Lancang River (also called the Mekong River outside China), Yangtze River, and Pearl River. Half a billion people live in the mid- and downstream areas of these river basins, where the above regions are vulnerable to hydrological extremes occurring in the upstream area [31]. In the eastern part of the study area, a large area of bare karst is developing. In low-permeability rocks surrounded by high-permeability rocks, water can easily infiltrate deep into the ground, resulting in surface water deficiency [32]. The study area is located between the eastern Himalayan syntaxis and the Sichuan Basin. Local stress distribution is also controlled by northward subduction of the Indian plate and the eastward subduction of the Burmese microplate [33]. According to crustal model from [34], the northwest crust is composed by quartz and diorite in the upper and the lower crust, while the rest contains granite and diabase in the upper and the lower crust, respectively [35]. With a complex geological background, the study area is also affected by frequent seismic and tectonic activities and other geological hazards. Such natural hazards resulted in the installation of continuous GPS observation sites in southwest China under the Crustal Movement Observation Network of China (CMONOC) since 2008 [36]. 

### 2.2. GPS Data

In total, there are 34 continuous GPS stations affiliated with CMONOC (Figure 1), spatially distributed mainly in Yunnan Province and partly in Sichuan and Guangxi Provinces; these GPS data could be accessed via http://www.cgps.ac.cn/. Majority of the continuous GPS sites in the study area have been built and initiated since the construction of CMONOC network that began in 2010, except KMIN and XIAG sites that have been in operation since 1999. Due to the debugging and adjustment of instruments in the incipient observation, data obtained before 2012 are of poor quality with variable trends and gross errors. Therefore, we ignored the GPS data before 2012, and used uniform data collected from all involved GPS sites between January 2012 and October 2014.

### 2.3. GRACE and GLDAS Data

TWS inferred from GRACE and GLDAS were both introduced as data constraints during the inversion process, and served as external data to verify and compare with the GPS-inferred TWS from the proposed methodology and the existing ones. Monthly TWS products, released by JPL and derived from GRACE RL05 GSM products, have been processed by applying a decorrelation filter and Gaussian filter with a radius of 300 km. They can be accessed at http://grace.jpl.nasa.gov, supported by the NASA MEASUREs Program [37,38,39]. The GLDAS-inferred TWS can be acquired by summing up soil water, canopy water, and snow water data layers. The monthly NOAH GLDAS data products used in this study are available at the Goddard Earth Sciences Data and Information Services Center [5]. To be consistent with GPS data span interval, the GRACE and GLDAS data from January 2012 to October 2014 were selected.

## 3. GPS Data Processing

In this section, we briefly summarize the pre- and post-processing of GPS time series to extract the annual signal. First, conventional GPS pre-processing steps were employed using GAMIT software. Second, different post-processing techniques were applied to: (1) eliminate non-hydrological signals contributing to the GPS VD through different modeled data; (2) suppress signal aliasing and common mode errors via spectral analysis and regional stacking filtering; and (3) extract the annual signal from the filtered GPS time series with the maximum likelihood estimation. 

### 3.1. GPS Data Preprocessing

Using the GAMIT software, we aligned the precise coordinates of the GPS sites to the International Terrestrial Reference Frame 2008 (ITRF2008) assisted by the International GNSS Service (IGS) stations with their positioning uncertainties constrained to 5 cm. During the pre-processing, it was necessary to correct various errors existing in positioning. We applied the following standard procedures to the GPS daily solutions: (1) precise ephemeris provided by IGS were adopted to fix satellite orbits; (2) the former three-order terms of the ionospheric delay were considered and corrected by GAMIT [40]; (3) tropospheric delay was jointly corrected by the global pressure and temperature (GPT) model and the Vienna mapping function 1 [41]; and (4) the receiver antenna center offsets were corrected by the antenna correction files from IGS. The gross errors were detected and eliminated beyond twice the range of the standard deviation.

### 3.2. Extraction of TWS Signal from GPS

To extract the displacement signal caused by TWS, non-hydrological sources contributing to the GPS VD must be eliminated. For this purpose, we adopted the FES2004 model provided by AVISO+ for the ocean tidal loading correction. The pole tide was eliminated by the elastic pole tide model from the International Earth Rotation Service (IERS) [42]. The non-tidal ocean loading, and the atmospheric loading were removed from GPS time series using the global geophysical fluid center data products (http://geophy.uni.lu/). 

### 3.3. Spectral Analysis and Filtering

To extract the annual signal from GPS, different frequencies and phases, including aliasing and draconitic effect, which leads to biased estimation of annual amplitude, have to be eliminated and mitigated. Therefore, it is necessary to examine the GPS time series in the spectral domain thereby filtering out disturbing signals.

Fast Fourier transformation (FFT) was applied to generate a power spectrum for each GPS time series (Figure 2). Dominant harmonics close to one cycle per year (cpy) are generally recognized as the annual signal. However, the period (averaged period of 318.3 days) deviates from the true annual period. GPS draconitic error should not account for this phenomenon, since it has a period of 351 days [43,44]. Previous researchers have found that noise caused by unmodeled geophysical processes and unidentified GPS technical errors can alias to the semiannual and annual signals, leading to a biased estimation of annual amplitude [45,46]. According to the Rayleigh criterion [47], harmonics with the frequency difference greater than 0.33 cpy can be resolved according to the sampling rate and length of the time series. We speculate that the signal aliasing results in a distorted annual signal. Examining the harmonics of 2nd and the 3rd dominant frequencies of each GPS site revealed that the oscillation period varies irregularly and deviates from 2 cpy severely (Figure 2). This indicates that the aforementioned errors may be propagated to the annual signal. Consequently, the harmonics of the 2nd and 3rd dominant frequencies were filtered out to avoid a noise signal aliasing. The filtered time series were retrieved back to time domain via the inverse FFT. 

For better demonstration of the effectiveness of this procedure, the spectra of the unfiltered and filtered time series are depicted in Figure 2. Before filtering, several GPS sites such as YNCX, YNMJ, YNML, exhibited a similar situation as YNLC did. The top peak deviated from the annual period of 365.25 days by up to 120 days. Other GPS sites would also exhibit a similar situation as these sites, although not as severe as they are (i.e., annual period deviation of 29.5 days). After filtering, the top peak is apparently closer to 1 cpy, particularly for the YNLC site. The mean deviation from the annual period decreases to 15.6 days, which can partly be ascribed to the GPS draconitic error. In addition, the energy of the annual signal becomes more apparent. This indicates that the spectral filtering can help restore a relatively good annual signal. 

### 3.4. Regional Stacking Filtering

Common mode error (CME) is a spatially-correlated error presented in regional GPS networks [48], which was first discovered in [49]. The elimination of CME can decorrelate the time series of different GPS sites [50] and enhance the signal-to-noise ratio [51]. Previous studies applied an empirical orthogonal function analysis [52] or a principal component analysis [14] to filter out the CME. In this study, we applied the regional stacking filtering technique proposed by [53] in the following steps: (1) select the daily solution of all GPS sites with common data time span; (2) calculate the residual series of each GPS site on the basis of Equation (1); and (3) average all the residual time series of the selected GPS sites to obtain the offset due to regional signal, followed by removing it from the original time series of all the GPS sites. Figure 3 illustrates that the GPS time series become less dispersed and exhibit more conspicuous annual variations and secular trend patterns, implying that the CME removal can distinguish the annual signal and improve the signal-to-noise ratio.

### 3.5. GPS Annual Signal Estimation

To extract the annual signal, a conventional trajectory model (Equation (1)) was adopted which comprises of periodic terms and linear trend term [54], given as:(1)$$y=a+b(t-t_{0})+\sum_{n=1}^{2}A_{n}\cos(\frac{2\pi }{T_{n}}(t-t_{0})+\varphi_{n})+\varepsilon$$
where $A_{n}$, $\varphi_{n}$, and $T_{n}$ are, respectively, the amplitude, phase, and period of periodic signals, ε is the noise term. Here, the origin time of observation, $t_{0}$, is set to January 1, 2012. The term *a* denotes the intercept and *b* denotes the GPS linear velocity. $T_{1}=1$ and $T_{2}=0.5$ represent the periods of the annual and semiannual signal terms, respectively. 

Least-squares analysis is typically applied to estimate the parameters in Equation (1). However, the application of least-squares adjustment will yield a biased estimation in the presence of colored noise (e.g., random walk and flicker noise) [55]. Therefore, we applied the maximum likelihood principle to estimate both, the periodic signals and noise components. To describe the stochastic process of the noise, the model of power-law noise and white noise combination was selected. A method based on a full covariance matrix was chosen for a likelihood computation and accomplished using the Hector software package [56]. 

A bedrock thermal expansion also contributes substantially to the annual amplitude. It will become larger as the annual variation of land surface temperature increases [57]. This annual variation can result in bedrock deformation up to about 0.5 mm in our study area [58], which in turn, causes an overestimation in TWS of about 1 cm, as tested. Therefore, we removed the effect of bedrock thermal expansion from the annual amplitude of the GPS according to a procedure used previously [58]. 

The result is summarized in Table 1. The mean GPS velocity is −0.55 mm per year, implying a slight submerged tendency in the study area from 2012 to 2014. With a mean value of 7.56 mm, the GPS height annual oscillation displays a general pattern; decreasing from southwest towards the northeast. According to the observation feature of the GPS sites, they are more sensitive to near-field loading. Thus, they can reflect a loading variation locally, which will also lead to the site-dependence distribution pattern. In addition, a different elastic response due to local crustal structure [35] or other unmodeled geophysical processes [59] can be a reason.

## 4. Inversion Methodology of TWS from GPS

GPS VD were inverted for TWS based on the elastic loading theory, which quantitatively describes crustal response to its surface loading [15]. This section illustrates the joint inversion of GPS VD with external GRACE and GLDAS TWS data via the Akaike’s Bayesian Information criterion (ABIC). ABIC will be described and employed to solve for the inversion problem from loading displacement to loading mass. 

### 4.1. Inversion Model

The inversion model is constructed based on elastic loading theory, which was developed based on the 1-D spherical Earth model. The Green function, relating a point mass, due to TWS, to the loading responses is presented by [15]:(2)$$G(\Theta )=\frac{a}{m_{e}}\sum_{n=0}^{\infty }h_{n}P_{n}(\cos\Theta ),$$
where *a* and *m_e_* are the radius and mass of the Earth, respectively, *h_n_* is the load Love number of the *n*-th degrees, $P_{n}(\cos\Theta )$ is the Legendre function of *n*-th degrees. The loading displacement at a point is the synthetic effect of all loading masses on the surface. The displacement contribution of a point mass is defined as the product of the Green function and a point mass in the following form:(3)du=G(Θ)dm,
where Θ is the angular distance between the point mass, *d_m_*, and the field point. Green’s function from [60] is adopted. The loading displacement at a point requires the surface integral of *d_u_*, which, in turn, sums up the contribution from all the loading masses on the surface. The loading masses outside the study area will also lead to loading displacement within the study area. Therefore, the integral region, equally divided into 1° × 1° grid, is extended with an extra 5° in each direction, as shown in Figure 4. Namely, the loading mass within the region 16.5°N ~ 34.5°N, 92.5°E ~ 110.5°E was considered. The loading masses outside this region were not considered due to their negligible contributions. 

Due to the deficiency of the GPS observations in our cases, totally 34 GPS sites, the number of observations is less than that of the parameters, totally 324. Thus, this is an underdetermined problem. However, only the TWS distribution within the study area is concerned. Namely, only 64 parameters, within the study area, is adopted. Here, Tikhonov regularization is adopted [61] and the Laplacian operator is selected as the Tikhonov matrix. Thus, this underdetermined problem takes the following form [24]:(4)$$\left\{ \begin{array}{l} Ah=u\\ \nabla^{2}h=0 \end{array}\right.,$$
where **h**, with the length of 324, denotes the column vector of the TWS annual amplitude of each grid to be estimated; **u,** with the length of 34, denotes vertical annual amplitude of each GPS site. The observation matrix, *A*, with the size of 34 × 324, in Equation (4) can be expressed as:(5)$$A=\left(\begin{array}{cccc} \iint_{\Omega_{1}}\rho_{w}G(\Theta_{1})dS & \iint_{\Omega_{2}}\rho_{w}G(\Theta_{1})dS & \cdots & \iint_{\Omega_{i}}\rho_{w}G(\Theta_{1})dS\\ \iint_{\Omega_{1}}\rho_{w}G(\Theta_{2})dS & \iint_{\Omega_{2}}\rho_{w}G(\Theta_{2})dS & \cdots & \iint_{\Omega_{i}}\rho_{w}G(\Theta_{2})dS\\ \vdots & \vdots & \ddots & \vdots \\ \iint_{\Omega_{1}}\rho_{w}G(\Theta_{j})dS & \iint_{\Omega_{2}}\rho_{w}G(\Theta_{j})dS & \cdots & \iint_{\Omega_{i}}\rho_{w}G(\Theta_{j})dS \end{array}\right),$$
where $\rho_{w}$ is the freshwater density, $\Omega_{i}$ is an *i*-th patch of integral surface, and $\Theta_{j}$ denotes the angular distance between a point mass and the *j*-th field point corresponding to the location of the GPS site. Each $\Omega_{i}$ is discretized into 0.025° × 0.025° to derive the surface element, dS, the area of mass grid. The Laplacian operator, $\nabla^{2}$, was replaced with the notation, *L*, for conciseness; thus, the regularized solution from Equation (4) takes the following form:(6)$$\hat{h}={\left(A^{T}A+kL^{T}L\right)}^{-1}A^{T}u$$
Note that a parameter, *k*, is included in the least-squares solution known as a smoothness factor, which controls the value difference between the centric grid and adjacent grids. The spatial distribution of TWS becomes more homogeneous with increasing value of *k*.

### 4.2. Joint Inversion of GPS VD and GRACE or GLDAS TWS

The spatial distribution of the GPS sites is uneven with more sites situated in the central part of the study area, and few sites in the northeast. Due to the absence of observations, the inversion result in the area with a sparse coverage by GPS sites probably deviates from a true situation and yields inferior inversion stability, resulting in reduced reliability. To remedy these situations, we introduced other observations at locations with a sparse coverage of GPS sites as data constraints for the joint inversion with GPS. This is equivalent to adding stochastic constraints to linear observation equations [62]. The TWS from GRACE and GLDAS were selected, owing to their common availability. 

The TWS from GRACE and GLDAS were not directly applied, but first converted to the hydrological loading following the concept of Equation (3), in which the complete formulation can be found in [60]. During the joint inversion, the role of the introduced data can be assumed as “pseudo GPS sites”, which merely record the crustal deformation exerted by the surface loading from TWS, without the contribution from other sources or noise disturbances. Observation data from these “pseudo GPS sites” can be obtained using GRACE-inferred or GLDAS-modeled TWS by the forward modeling. Equation (4) can then be extended to:(7)$$\left\{ \begin{array}{l} Ah=u\\ Lh=0\\ A^{\prime }h=u^{\prime } \end{array}\right.$$
where $u^{\prime }$ denotes the synthetic VD at specific site locations, and $A^{\prime }$ is the corresponding observation matrix with size depending on the number of synthetic GPS locations from the supplementary GRACE or GLDAS TWS data. The solution of Equation (7) can be expressed as
(8)$$\hat{h}={\left(A^{T}A+kL^{T}L+{A^{\prime }}^{T}A^{\prime }\right)}^{-1}(A^{T}u+{A^{\prime }}^{T}u^{\prime })$$

The selection of location and number of the “pseudo GPS sites” is of vital importance to the inversion quality. During the location selection process of the pseudo GPS sites, the following two semi-empirical steps were proposed: (1) one pseudo GPS site will be placed on one patch each time and would participate in the joint inversion, and (2) this result will be compared with the TWS from GRACE or GLDAS and the mean bias will be found between them. The location with a smaller mean bias will be primarily chosen for the joint inversion. 

Besides this selection process, the density of the GPS sites should also be taken into consideration. Locations with low coverage of GPS sites should be equipped with “pseudo GPS sites”. GPS is more sensitive to variations of local loading, which probably affects the result of inversion. To pose a minimum influence, the pseudo GPS sites should be as few as possible so that the GPS can play a dominant role during the inversion. 

### 4.3. Akaike’s Bayesian Information Criterion (ABIC)

ABIC is an entropy method that combines the Akaike Information Criterion and the Bayesian theorem. The Bayesian theorem can describe the probability distribution of TWS at the study area for given observations (i.e., GPS, GRACE, or GLDAS) and prior information (i.e., Laplacian constraint). ABIC can select a model based on the entropy maximization principle, presented by [63,64]:(9)$$ABIC(\alpha^{2})=n\log(s(h^{*}))-\log\left|\alpha^{2}L^{T}L\right|+\log\left|A^{T}A+\alpha^{2}L^{T}L\right|+C^{\prime }$$
where *n* is the number of observations, and $\alpha^{2}$ is the parameter reflecting the variance ratio between the observation and constraint equations. $C^{\prime }$ is a constant term, which does not influence the selection of the parameter corresponding to a minimal ABIC. Note that the covariance matrix of the observation vector is omitted, while it was taken into consideration in [63,64]. The weighting for each observation is set as equal. The covariance matrix, $E(\sigma^{2})$, then becomes the identity matrix. s(h) is the Lagrangian target function that optimizes the modeling error and roughness as follows:(10)$$s(h)={\left(u-Ah\right)}^{T}(u-Ah)+\alpha^{2}h^{T}L^{T}Lh$$
Equation (10) is similar to the metric in the error space. However, the observations have equal weightings, and thus amounts to the identity matrix. $h^{*}$ is the optimal solution when optimal parameter $\alpha^{2}$ is assumed, which can be solved by minimizing s(h) when setting ∂s/∂h=0, such that:(11)$$h^{*}={\left(A^{T}A+\alpha^{2}L^{T}L\right)}^{-1}A^{T}u.$$
Indeed, it is the least-squares solution of Equation (4) and thus identical to Equation (6). Compared with Equation (6), $\alpha^{2}$ is the smoothness factor in this case. As for the ABIC extension to the case with GLDAS or GRACE constraints, we do not weigh between GPS and additional constraint; Consequently, the ABIC expression is simplified, which can be achieved by replacing $u_{1}={\left(\begin{array}{cc} u & u^{\prime } \end{array}\right)}^{T}$ and $A_{1}={\left(\begin{array}{cc} A & A^{\prime } \end{array}\right)}^{T}$.

## 5. Results

To verify the feasibility of constraint strategy and to make a quantitative comparison, TR and HVCE, which are the only two methods that have been applied in recent literature, were employed and subsequently compared with the result without constraint. For a detailed description of TR and HVCE, we refer readers to the studies of [16,17,24].

### 5.1. TWS Inferred from GPS without Constraint Points

Regardless of inversion methods, the TWS inferred from GPS exhibits a general pattern of the north-south stripe with the TWS magnitude in the western part larger than that in the eastern part (first column of Figure 5). The TWS is distributed homogeneously on the same meridian, but diminishes rapidly along the parallel, particularly when inverted with HVCE. The TWS inverted by HVCE is 28.5% and 21.4% larger than the one inverted with TR and ABIC, respectively, in the western part of the study area, which also shows the largest difference up to 20 cm compared with the result in the eastern part. The difference between the results inverted with TR and ABIC is small with a mean difference of 0.1 cm, but displays relatively large differences in the western and eastern edges.

### 5.2. TWS Inferred from GPS with Constraint Points

As shown in the Figure 5b,e,h and Figure 5c,f,i, three to four constraint points are involved in the joint inversion, mostly distributed near the edge of the study area. Compared with the result without constraint points, TWS with constraints shows a general increasing pattern from northwest to southeast. The TWS distributed along the Lancang River is larger than those along the other rivers. The mean difference between the TWS with the GLDAS constraint and that with GRACE constraint is 0.94 cm and the former result is smoother. The result inverted with the GRACE constraint is generally larger than the GLDAS result in the west while it is smaller in the east. Regarding the inversion methods, the regional characteristics of TWS with HVCE or ABIC are more distinguishable than the one inverted with TR, whose result is also the smoothest.

In the mutual comparison among the GPS-inverted TWS, it is obvious that the one inverted with HVCE without constraint points shows the largest bias compared with other inversion results. We eliminate this solution from the solution set. The spatial distribution of standard deviation of the GPS-inverted solutions is calculated (Figure 6). With a mean standard deviation of 1.28 cm, the distribution of the GPS sites is correlated with the standard deviation. The northwest part of our study area, with a scarce distribution of GPS sites, exhibits large biases among different solutions, indicating the importance of a uniform data constraint condition. The standard deviations in the central and the eastern parts of the study area are evidently smaller than that in the adjacent area, indicating the reliability of the result in this region.

## 6. Discussion

The TWS data inverted from the GPS with three kinds of inversion strategies under the Laplacian condition are without data constraint, with GRACE data constraint, and with GLDAS data constraint, which are abbreviated as no constraint, GRACE constraint and GLDAS constraint, respectively. It is necessary to analyze the features and potential discrepancies of every constraint condition and inversion method and to evaluate the relatively optimal solution to determine the suitable inversion method and constraint condition. Here, the solutions are discussed and evaluated based on three aspects: model fitness, inversion process, self-comparison and comparison with other observations.

### 6.1. Model Fitness

Model fitness can be reflected from the root-mean-square residuals (RMSR) of the observation equation (RMSR^1^) and the constraint equation (RMSR^2^), as shown in Table 2. It is evident that additional data constraints are not favored by the model since RMSR^1^ is larger than it is in the case without constraint. Due to the incompatibility between the GPS and the additional data constraints, extraneous errors are observed, which decrease the model fitness. This incompatibility depends on the bias between the GPS-derived and GRACE- or GLDAS-derived VD, which will increase RMSR^1^ and RMSR^2^ if the bias increases. The bias between GPS VD and GRACE-derived VD is smaller than the bias between the GPS and GLDAS-derived one. The GRACE-inferred TWS is larger than that of GLDAS in most area, as displayed in Figure 5j,k. Thus, both RMSR^1^ and RMSR^2^ for the GRACE constraint is smaller than those with the GLDAS constraint. The inversion with GRACE constraint has better model fitness performance and more efficient constraint effect than the GLDAS constraint, which demonstrates that the GRACE constraint is a more appropriate constraint source.

As for the model fitness comparison among inversion methods, the result inverted with HVCE yields the best model fitness. ABIC comes second, while TWS inverted by TR has the largest RMSR^1^. This implies that HVCE is superior to ABIC and TR in terms of improving the model fitness. Each method has different target functions that result in different model fitness values. HVCE can balance the variance of each equation to re-weigh different datasets iteratively, thereby reducing the misfit of observation and constraint equations [65]. ABIC pays additional attention to the information entropy of the model, in addition to the model residuals and constraint residuals [27,64]. Furthermore, TR sets equal weights for the model misfit and roughness in the error space and reaches the tradeoff among different datasets with an equal distance, which cannot effectively suppress the error from additional constraint according to our result. 

Compared with the cases with HVCE and TR, ABIC can reduce the loss in the model fitness to a minimum. Furthermore, it yields a stable model fitness performance regardless of the constraint condition, which results from the proximity of the smoothness factor to each other in every constraint condition, as displayed in Figure 7b and Table 3. As for TR, roughness varies significantly as smoothness factor changes, as shown in Figure 7a. However, due to the existence of parameter *α*, as displayed in Equation (11), the roughness has a negligible impact on the error metric of inversion, *s*(*h*). Basically, the ABIC target function accounts less for the roughness and weighs more for the constraint error and the model fitness, which is more stable as *k* changes. Therefore, the inversion using ABIC can generate relatively similar results with different constraints.

### 6.2. Inversion Process

As for the inversion process, the spatial distribution of mean uncertainty and problems associated with the inversion process (e.g., sensitivity to the initial value) are evaluated, which helped us understand the reliability of the inverted solutions.

Due to unawareness of the spatial distribution of uncertainty, inversion uncertainty is estimated with the bootstrapping method. The principle of the bootstrapping method is that one GPS site is eliminated and the rest participate in the inversion as usual. This site is then included again in the next time and another GPS site is eliminated. This process continues until all GPS sites have been eliminated for one time and a solution set is acquired, whose standard deviation is the uncertainty. 

As displayed in Figure 8, the uncertainty distribution is highly correlated with the distribution of the GPS sites and constraint points, particularly for the case with the constraint. Besides, the uncertainty in the margin of the study area is conspicuously, relatively large, indicating relatively poor inversion performance. With the introduction of a few constraint points, the uncertainty is reduced by 42% on average when compared with the cases without constraint, particularly in the boundary area. 

Different constraint conditions have various impacts on the uncertainty of the inversion results. The GRACE or GLDAS constraints can reduce the inversion uncertainty (i.e., mean uncertainty) by 51% and 66% in the case of HVCE and TR, respectively, but only by 9% in the case of ABIC (Table 2). This implies that the introduction of constraint points can mitigate the deterioration of the inversion performance due to uneven distribution of the GPS sites, thereby improving the inversion stability.

In the iteration process of HVCE, the variance of unit weight is considerably sensitive to a given initial value, which is prone to be divergent as the error accumulation and propagation for an inappropriate initial value. The selection of the initial value requires repeating trials. This selection process does not affect the inversion result as long as it converges. The initial value is prescribed as 1 × 10^−4^, which is suitable for both with or without constraint. However, if this technique is extended to weight determination among various observations and constrain conditions, more onerous trials are necessary for the appropriate initial value. Therefore, the requirement for the prudently-assigned initial value limits the feasibility and extension capability of this inversion method.

### 6.3. Comparison with Other Observations

Comparison with the results from other observations is indispensable for the validation of our result, and it demonstrates its reliability. Here, our result is compared with the result from GRACE and GLDAS. 

Comparing Figure 5j,k, it is not hard to discover that GLDAS-inferred TWS is smaller than that from GRACE, and the former one has more distinct regional characteristics than the smoother GRACE-inferred TWS. However, when compared with the above two results, the GPS-inferred TWS is universally the largest regardless of the constraint source and inversion method, particularly for the case without constraint. The inversion result in the northwest area contributes the most to the bias between the GPS-inferred results and other observations, where the inversion result has the largest difference among different GPS-inferred methods, as seen in Figure 6. The GPS-inferred TWS in the northwest area is 8 cm larger in the case of the constraint applied and 14 cm larger at least in the case without constraint, which is the major bias contribution. 

Other researchers have reported similar problems that the GPS-inferred TWS is generally larger than other observations [16,17,18]. Fu et al. [17] discovered that GPS-inferred TWS is larger than the GRACE one, with the root-mean-square reduction of 62.3%, implying the remaining discrepancy of 37.7%. To compare with his results, the ratio of root-mean-square difference between GPS-inferred TWS and GRACE-inferred TWS to the mean of GRACE-inferred TWS, serving as the discrepancy metric, is calculated in the sixth column of Table 2, denoting as Discrepancy Ratio^1^ term. It is obvious that our discrepancy estimates are smaller than that reported by Fu et al. [17], in particular the case when “pseudo GPS sites” is introduced. Note that GRACE suffers from spatial and temporal aliasing errors [6,7], and leakage effects, while GLDAS merely considers snow water, canopy water, soil water within 2 meters. Therefore, it is hard to judge whether the GPS-inferred TWS is overestimated due to the sensitivity of different observed and modeled data.

To evaluate the bias more objectively, an internal (i.e., GPS-inferred TWS compared with constraint observation) and an external (i.e., GPS-inferred TWS compared with non-constraint observation) assessment are considered. The application of GRACE or GLDAS constraint reduces the bias by 45% and 36% for the internal assessment, whereas the bias is diminished by 32% and 49% for the external assessment. 

In terms of the spatial distribution pattern, the GPS-inferred TWS with constraint points is more similar to the TWS from GRACE or GLDAS than the one without constraint. This can also be reflected in the mean bias term (Table 2). It demonstrates that the application of GRACE or GLDAS constraint can effectively reduce the bias against other observations and improve the inversion result. 

In terms of the inversion method, the application of GRACE or GLDAS constraint can reduce the bias by 52%, 36%, and 34%, respectively, for HVCE, TR and ABIC in the external assessment. The TWS inverted with HVCE and no constraint (Figure 5a) shows the largest bias in comparison with the result from both GRACE and GLDAS or the GPS-inferred results with another inversion method. In spite of the good model fitness, this result is not considered reliable because it fails to verify with other observations and possesses a relatively large uncertainty. With the introduction of GRACE or GLDAS constraints, the TWS inverted with TR exhibits small differences with the TWS inferred from GRACE or GLDAS, but deteriorates the model fitness significantly. The solution inverted with ABIC yields a steady performance in terms of bias comparison, uncertainty, and model fitness, with or without constraint. 

## 7. Conclusions

In this study, we inverted TWS in southwest China from continuous GPS stations affiliated to CMONOC based on the Akaike’s Bayesian Information Criterion. To mitigate the deterioration of inversion quality owing to the uneven or potentially sparse distribution of the GPS sites, GRACE or GLDAS data (serving as pseudo-GPS sites) were minimally introduced in the joint inversion. To verify the feasibility of our constraint strategy, other GPS-inferred methods, TR and HVCE, were implemented with GRACE or GLDAS constraints. Our results revealed that the introduction of GRACE or GLDAS constraint points can improve the inversion results by refining the spatial distribution, reducing the inversion uncertainty and bias with other observations, despite the deterioration of the model fitness. We further analyzed the advantages and disadvantages of each GPS-inferred method. Our constraint strategy improved the inversion performances of HVCE, TR, and ABIC dramatically. As for HVCE, apparently, the inversion performance can be enhanced, although it suffers from sensitivity to the initial value. As for TR, the application of constraints can improve inversion stability evidently and reduce the bias compared with other observations, though, at the expense of the model fitness quality. With the minimal deterioration of the model fitness, ABIC yields stable model fitness performance regardless of the constraint condition. Based on the comprehensive consideration, ABIC appears to be an appropriate inversion method for TWS inversion inferred from GPS.

## Figures and Tables

**Figure 1 entropy-21-00664-f001:**
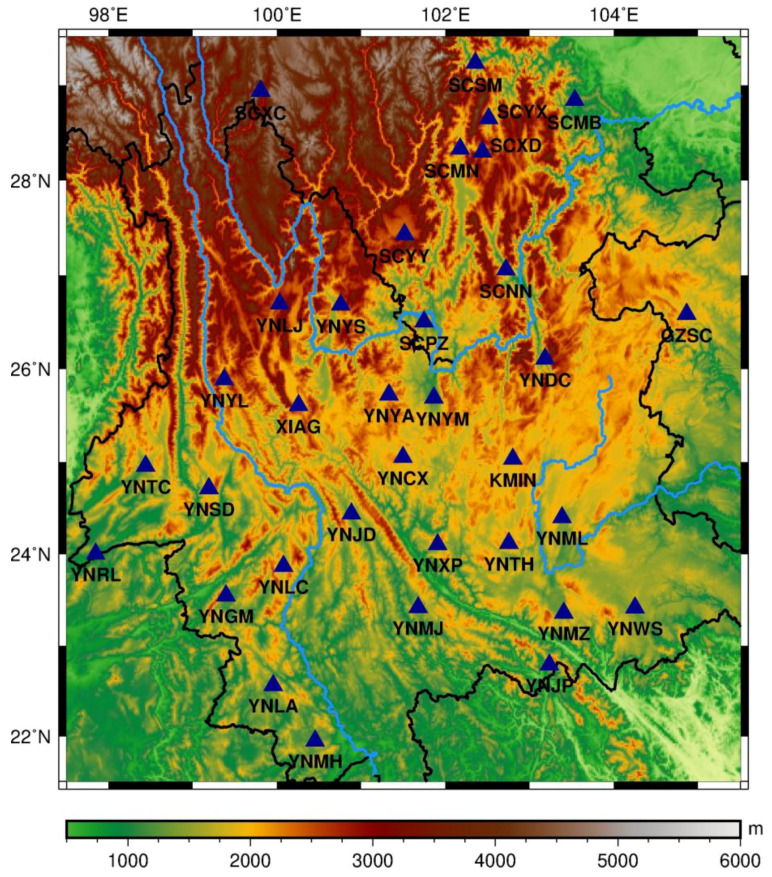
Topography of southwest China. Continuous GPS site locations (in red triangles).

**Figure 2 entropy-21-00664-f002:**
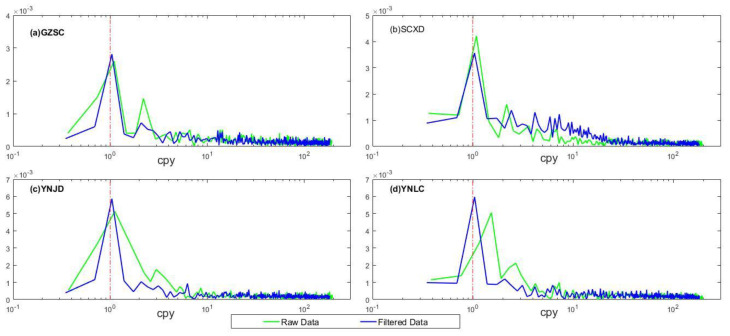
The unprocessed GPS spectra (green line) and the processed one with the 2nd and 3rd dominant frequencies filtered (blue line) in spectral domain, x-label denotes the cycle per year.

**Figure 3 entropy-21-00664-f003:**
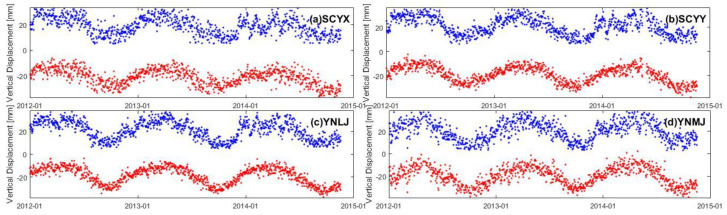
Comparison between GPS data before (blue scatter) and after (red scatter) applying regional stacking filter. The annotation in the top-right corner is the GPS site name.

**Figure 4 entropy-21-00664-f004:**
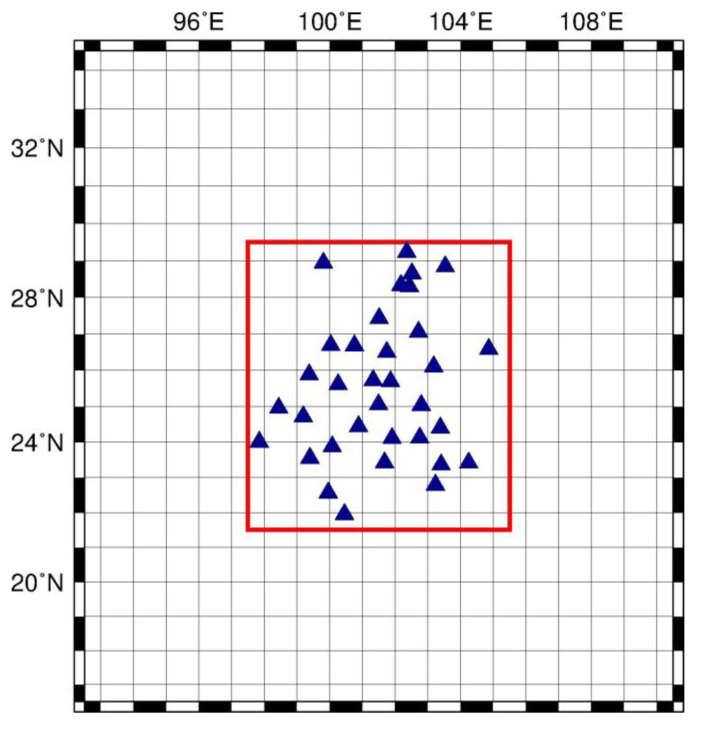
Sketch of integral surface extension and grid partitioning. The study region is marked inside red box. GPS sites are marked as blue triangle.

**Figure 5 entropy-21-00664-f005:**
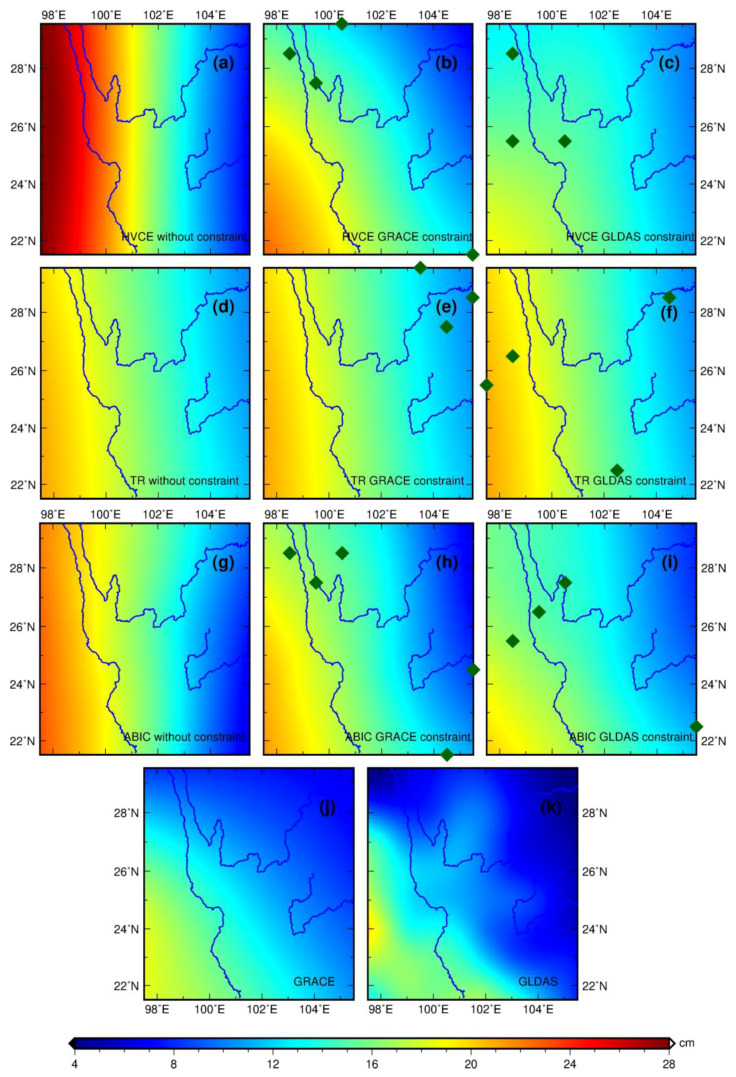
The TWS inferred from GRACE, GLDAS, and GPS with different inversion methods and constraint conditions. Figures of the first, second, and third row denote inversion results from HVCE (**a**–**c**), TR (**d**–**f**), and ABIC (**g**–**i**), respectively. The figures in the fourth row are the TWS inferred from GRACE (**j**) and GLDAS (**k**). Figures of the first, second and third columns denote inversion results without constraint (**a**,**d**,**g**), with GRACE constraint (**b**,**e**,**h**) and with GLDAS constraint (**c**,**f**,**i**), respectively. The dark-green diamonds denote the location of “pseudo GPS sites”.

**Figure 6 entropy-21-00664-f006:**
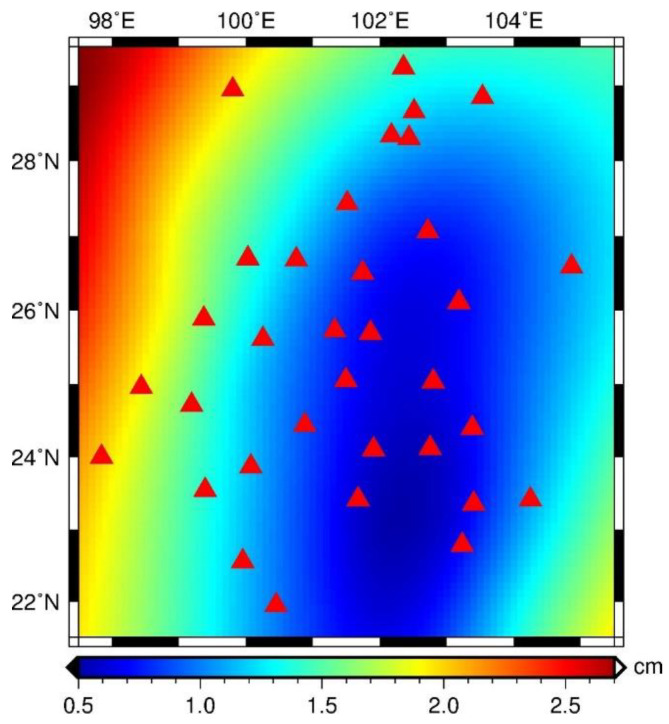
The standard deviation distribution of the solutions inverted with different methods and constraint conditions. The red triangles denote the GPS sites.

**Figure 7 entropy-21-00664-f007:**
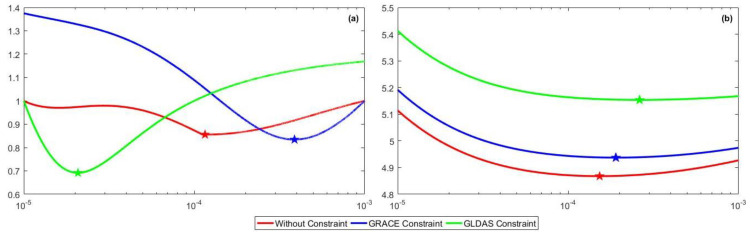
The process of smoothness factor selection. The *x*-axis denotes the smoothness factor. (**a**,**b**) denote the selection processes of TR and ABIC, respectively. The minimum point of each curve is marked by a star symbol.

**Figure 8 entropy-21-00664-f008:**
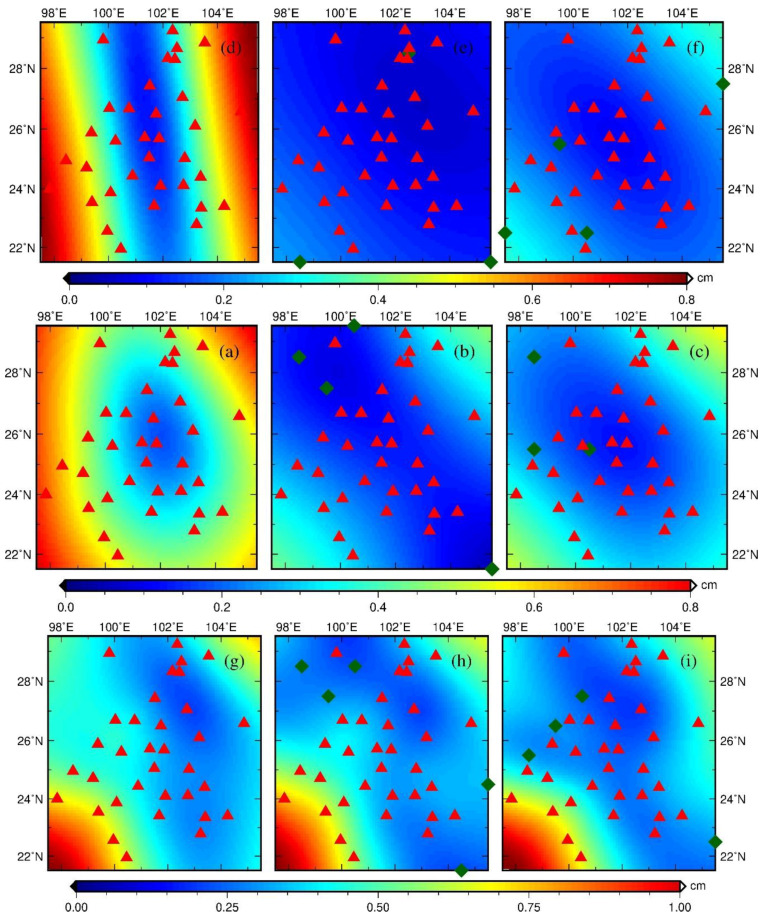
The spatial distribution of GPS-inferred TWS uncertainty. The annotation in the top right corner of each figure has an identical implication to the one in Figure 5. The red triangles and dark-green diamonds denote the GPS sites and constraint points, respectively.

**Table 1 entropy-21-00664-t001:** GPS velocity, annual amplitude, and corresponding standard deviation.

GPS Site	GPS Velocity (mm/year)	Annual Amplitude (mm)	GPS Site	GPS Velocity (mm/year)	Annual Amplitude (mm)
GZSC	−0.12 ± 0.35	4.87 ± 0.32	YNLA	−3.25 ± 1.28	10.94 ± 0.78
KMIN	2.09 ± 1.40	5.00 ± 0.80	YNLC	−1.09 ± 0.46	10.60 ± 0.43
SCMB	−1.10 ± 0.53	3.42 ± 0.43	YNLJ	−1.32 ± 0.34	8.77 ± 0.28
SCMN	−1.00 ± 0.35	7.34 ± 0.32	YNMH	−1.04 ± 2.19	6.71 ± 1.13
SCNN	1.93 ± 2.47	8.02 ± 1.25	YNMJ	0.88 ± 0.45	9.71 ± 0.41
SCPZ	−0.74 ± 0.22	7.92 ± 0.22	YNML	−1.34 ± 0.43	6.14 ± 0.35
SCSM	−0.64 ± 0.31	8.65 ± 0.30	YNMZ	−0.67 ± 0.33	7.77 ± 0.33
SCXC	0.75 ± 0.36	7.82 ± 0.29	YNRL	0.70 ± 0.71	10.76 ± 0.47
SCXD	−0.13 ± 0.32	8.08 ± 0.30	YNSD	−0.66 ± 0.38	10.39 ± 0.36
SCYX	−1.66 ± 0.28	6.25 ± 0.28	YNTC	−0.25 ± 0.69	10.50 ± 0.52
SCYY	−1.35 ± 0.23	7.84 ± 0.23	YNTH	0.33 ± 0.39	6.72 ± 0.36
XIAG	−1.28 ± 0.57	3.60 ± 0.52	YNWS	1.63 ± 0.64	5.41 ± 0.41
YNCX	−0.65 ± 0.39	7.39 ± 0.35	YNXP	−1.52 ± 0.55	4.83 ± 0.47
YNDC	−0.81 ± 0.47	6.32 ± 0.44	YNYA	−0.31 ± 0.29	8.46 ± 0.28
YNGM	−0.70 ± 1.04	2.44 ± 0.82	YNYL	−4.43 ± 1.28	8.90 ± 0.86
YNJD	0.58 ± 0.40	10.37 ± 0.39	YNYM	−0.18 ± 0.38	8.71 ± 0.35
YNJP	−1.06 ± 0.57	6.70 ± 0.46	YNYS	−0.45 ± 0.27	9.74 ± 0.27
			Mean	−0.55	7.56

**Table 2 entropy-21-00664-t002:** Statistical performance of the TWS inferred from the GPS with different constraint source and inversion methods. Mean Bias^1^ and Mean Bias^2^ denote the difference between the TWS inferred from GPS and TWS from GRACE and GLDAS, respectively. RMSR^1^ and RMSR^2^ mean the root mean square error of observation and constraint equation respectively. Discrepancy Ratio^1^ and Discrepancy Ratio^2^ denote the ratio of root mean squared difference between GPS-inferred TWS and GRACE-inferred TWS to mean GRACE-inferred TWS, or to mean GLDAS-inferred TWS, respectively.

	Mean Bias^1^ (cm)	Mean Bias^2^ (cm)	Mean Uncertainty (cm)	RMSR^1^ (cm)	RMSR^2^ (cm)	Discrepancy Ratio^1^	Discrepancy Ratio^2^
HVCE without constraint	6.24	7.49	0.46	1.08		0.54	0.74
HVCE GRACE constraint	3.00	4.50	0.20	1.52	0.68	0.26	0.44
HVCE GLDAS constraint	2.28	3.79	0.26	1.67	3.01	0.20	0.37
TR without constraint	4.11	5.62	0.42	1.44		0.35	0.55
TR GRACE constraint	2.08	3.61	0.11	2.04	0.74	0.18	0.36
TR GLDAS constraint	2.58	4.05	0.18	3.59	1.57	0.22	0.40
ABIC without constraint	3.99	5.41	0.43	1.90		0.34	0.53
ABIC GRACE constraint	2.69	4.18	0.39	1.94	1.03	0.23	0.41
ABIC GLDAS constraint	2.18	3.68	0.39	2.00	2.80	0.19	0.36

**Table 3 entropy-21-00664-t003:** Smoothness factor *k* of different inversion methods with various constraints.

	HVCE	TR	ABIC
Without Constraint	8.42 × 10^−4^	1.16 × 10^−4^	1.53 × 10^−4^
GRACE Constraint	8.61× 10^−4^	3.88 × 10^−4^	1.91 × 10^−4^
GLDAS Constraint	3.49 × 10^−4^	2.08 × 10^−4^	2.62 × 10^−4^

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
