# Peer review of "Akaike’s Bayesian Information Criterion for the Joint Inversion of Terrestrial Water Storage Using GPS Vertical Displacements, GRACE and GLDAS in Southwest China"

_entropy, 2019, doi:10.3390/e21070664_

Round 1

Reviewer 1 Report

GPS measurements are used in many areas of science. They have become a very valuable tool. I am glad that the presented work does not try to analyze the capabilities of GPS system, but to extract useful information from available measurements and use it for a well-defined research problem. A very interesting idea.

Author Response

Thank you.

Reviewer 2 Report

See the attached file. Generally, the inversion model is not quite clearly described.

Author Response

Please read the PDF for details.

Comments and Suggestions for Authors

See the attached file. Generally, the inversion model is not quite clearly described.
Questions and remarks to the paper

Akaike’s Bayesian Information Criterion for the Joint Inversion (…)” by Yongxin Liu et al.

1. General

a. Why is it so important to obtain TWS without the GRACE nor GLDAS data? Perhaps all available data should be taken to increase the TWS resolution?

Response: GPS, serving as an independent observation technique for TWS inversion, is under development stage. Many problems is pending to be solved, such as inversion algorithm and its performance analysis. However, only few researchers have studied TWS inversion methodologies from GPS. Therefore, we focus more on TWS inversion from GPS in this manuscript, while GRACE or GLDAS data are served as auxiliary information. The data fusion of GPS, GLDAS, GRACE to retrieve higher TWS resolution is a novel idea for further study, such as data fusion for resolution enhancement.

b. Some sentences should be added about the elastic loading theory, both generally and in the context of the area studied.

Response: Yes, you are right. We added more introduction about elastic loading theory into Section 4.1 in the revised manuscript as below:

"The inversion model is constructed based on elastic loading theory, which was developed based on the 1-D spherical Earth model. The Green function, relating a point mass, due to TWS, to the loading responses is presented by [15]

,

(2)

where  and  are radius and mass of the Earth, respectively,  is the load Love number of the nth degrees,  is the Legendre function of nth degrees."

[15]. Farrell, W. Deformation of the Earth by surface loads. Reviews of Geophysics 1972, 10, 761-797.

We supplemented geological background of the study area into Section 2.1 in the revised manuscript as below:

"The study area is located between the eastern Himalayan syntaxis and the Sichuan Basin. Local stress distribution is also controlled by northward subduction of the Indian plate and the eastward subduction of the Burmese microplate [33]. According to crustal model from [34], the northwest crust is composed by Quartz and Dioritein in the upper and the lower crust, while the rest contains Granite and Diabase in the upper and the lower crust, respectively [35]."

[33]. Yin, A.; Harrison, T.M. Geologic Evolution of the Himalayan-Tibetan Orogen. Annual Review of Earth and Planetary Sciences 2000, 28, 211-280, doi:10.1146/annurev.earth.28.1.211.

[34].Tesauro, M.; Audet, P.; Kaban, M.K.; Bürgmann, R.; Cloetingh, S. The effective elastic thickness of the continental lithosphere: Comparison between rheological and inverse approaches. Geochemistry, Geophysics, Geosystems 2012, 13, doi:10.1029/2012gc004162.

[35]. Dill, R.; Klemann, V.; Martinec, Z.; Tesauro, M. Applying local Green's functions to study the influence of the crustal structure on hydrological loading displacements. Journal of Geodynamics 2015, 88, 14-22.

c. And the area itself should also be better described in the context of the elastic loading theory – its geological structure, what is the effect on the elastic response of this structure to a loading displacement, how the structure is taken into consideration in the approach admitted, etc.

Response: We responded your first question in question 1 (b) above. The elastic loading theory is developed on the one dimensional spherical Earth model. Therefore, it does not consider the heterogeneity of crustal composition in the horizontal direction. So far, we did not find any research that geological structure, such as faults, has a substantial influence on annual TWS loading in this study region. We can only assume the elastic response properties in this region. In case you are asking the impact of crustal composition to loading displacement due to the difference of the elastic parameter, this can be represented by a local Green’s function (Dill et al., 2015), which should be a good idea for the potential improvement of our result in the future.

Dill, R.; Klemann, V.; Martinec, Z.; Tesauro, M. Applying local Green's functions to study the influence of the crustal structure on hydrological loading displacements. Journal of Geodynamics 2015, 88, 14-22.

d. Was such a short observation period (only 3 years) adopted on purpose? At least theyears 2015 and 2016 could be also admitted to the computations and analyses.

Response: Since December 2014, GRACE data products have become less reliable and feasible due to equipment maintenance and orbit error brought by excessive service, which can refer to https://isdc.gfz-potsdam.de/grace-isdc/older-news/ for more detailed information. GRACE data products are important for inversion constraint and inversion result validation. Therefore, we decided to adopt the data products only from 2012 to 2014.

2. GPS

a. What positioning technique was applied? Was it PPP or relative static?

Responding: It was a network solution, assisted by 24 International GNSS Service (IGS) stations surrounding China.

b. The explicit information on GPS positioning final accuracy must be given and shortly discussed; it is very important, in the light of “(…) bedrock deformation up to about 0.5mm in our study area [41], which in turn, causes an overestimation in TWS of about 1 cm(…), l. 168/169

Responding: You are right. We apologize for the misplacement of “bedrock thermal expansion correction” into Section 3.2, which should be placed at Section 3.5, line 233-238 according to our processing procedure. Therefore, it might lead to misunderstanding. The correction of bedrock thermal expansion is applied to annual amplitude of GPS vertical displacement time series. We have given the standard deviation of GPS annual amplitude in Table 1, the input of inversion model, which can serve as an indicator of GPS final accuracy. The GPS accuracy varies from sites to sites. Therefore, bootstrapping method is adopted to evaluate the inversion uncertainty of GPS-inferred TWS, etc.

c. Which of GPS results will be used in equations (3) and (6) as an input? Annual amplitude? Vertical displacement (if so - what it was referred to)? Velocity? It should be stated explicitly, and this chosen result and its accuracy estimation should be described in more details

Response: Annual amplitude is the input of equation (3) and (6). We have described it in line 281.

“…, u denotes vertical annual amplitude of each GPS site, with the length of 34.”

d. Is the linear velocity of GPS sites obtained from (1) taken into consideration in further computations?

Response: No, it was not considered in further steps.

3. Inversion methodology

a. Line 260: deficiency – give the exact number of observations and parameters

Response: We supplemented the exact number of observations and parameters in the revised manuscript as shown below:

"Therefore, the integral region, equally divided into 1° × 1° grid, is extended with an extra 5° in each direction, as shown in Figure 4. Namely, the loading mass within the region 16.5°N ~ 34.5°N, 92.5°E ~ 110.5°E was considered. The loading masses outside this region were not considered due to their negligible contributions.

Figure 4 Sketch of integral surface extension and grid partitioning. The study region is marked inside red box. GPS sites are marked as blue triangle.

Due to the deficiency of the GPS observations in our cases, totally 34 GPS sites, the number of observations is less than that of the parameters, totally 324. Thus, this is an underdetermined problem. However, only the TWS distribution within the study area is concerned. Namely, only 64 parameters, within the study area, is adopted. Here, Tikhonov regularization is adopted [61] and the Laplacian operator is selected as the Tikhonov matrix. Thus, this underdetermined problem takes the following form [24]

,

(4)

where h ,with the length of 324, denotes the column vector of the TWS annual amplitude of each grid to be estimated; u, with the length of 34, denotes vertical annual amplitude of each GPS site. The observation matrix, A, with the size of 34 × 324, in Equation (4) can be expressed as....."

b. More detailed description of vector h must be provided, in the context of equations (3)and (6). Vector u contains the vertical annual amplitude of each GPS site (l. 265), then his the amplitude of TWS annual changes? One per each dS for the whole 3-year period? Or a total change of TWS on each dS? State explicitly what is obtained from the inversion. It should be explained clearly.

Response: Please read 3(a) above. Vector h denotes the column vector of the TWS annual amplitude of each grid to be estimated. It has same meaning in equations (3) and (6). It represents the annual amplitude of TWS from 2012 to 2014.

c. In eq. (4) there occur dS (surface elements): how the area was divided into dSs? How many of them is obtained? Are they simply 1 0 x1 0 areas? How big they are? Can they be regarded as homogeneous in the context of water conditions and geological structure? If the division is different from 1 0 x1 0 then it must be shown in a figure (perhaps in Fig. 1)

Response: Each  is discretized into 0.025° × 0.025° to derive the surface element, , the area of mass grid. is an ith patch of integral surface, namely the 1° × 1° grid. According to elastic loading theory, the crust is considered as homogeneous in the horizontal direction. Therefore, we divided them equally. The extension of integral surface and the integral surface discretized into 1° × 1° grid is plotted in Figure 4 of the revised manuscript above.

" ...where  is the freshwater density,  is an ith patch of integral surface, and  denotes the angular distance between a point mass and the jth field point corresponding to the location of the GPS site. Each  is discretized into 0.025° × 0.025° to derive the surface element, , the area of mass grid...."

d. Dimensions of the matrix A – should be given explicitly, for all the 3 cases: without constraints under (3), and with Grace and Gldas constraints under (6)

Response: Please read 3(a) above. Thank you for reminding. The dimensions of A have been given explicitly in line 282, as shown below:

h, with the length of 324, denotes the column vector of the TWS annual amplitude of each grid to be estimated; u, with the length of 34, denotes vertical annual amplitude of each GPS site. The observation matrix, A, with the size of 34 × 324 …

The dimensions of h and u is also given, so the reader can deduct the dimensions of A easier. The dimensions of the matrix A do not change as the constraint conditions changes, because the number of GPS sites and gridding partitioning do not change.

e. Taking rms as accuracy indicator in case of under-determinated problem is a little hazardious; For a problem with equal number of parameters and observations it would be zero. What is in the case of under-determinated problem? At least a short discussion is required, augmented with literature citations

Response: It is not RMS error with respect to the true value. Actually, RMSR1 and RMSR2 are the root mean square residuals of observation equations and constraint equations, respectively, after the least-squares adjustment process. They are, in essence, variance factors, adopted for model fitness (internal) assessment, rather than for external accuracy assessment. We did a detailed discussion in Sector 6.1 about RMSR variation with respect to observation equation and constraint equation from the aspect of constraint source and inversion methods.

f. Generally, is comparing results constrained with Grace to Grace TWS correct? Isn’t it obvious that they will be closer to Grace? Can then Grace be called “other” observation?

Response: To a large extent, yes, because only a few GRACE constraint pseudo station locations are added. In the new Figure 5 in the revised manuscript, we indicates the added constraint locations. To compare the results more objectively, in Section 6.3, we have compared from two aspects, internal aspect and external aspect. For instance, if GRACE constraint is adopted, we can compare it with GLDAS-inferred TWS as the external aspect and compare it with GRACE-inferred TWS as the internal aspect. We have clarified it in the line 497-501 as below:

"To evaluate the bias more objectively, an internal (i.e., GPS-inferred TWS compared with constraint observation) and an external (i.e., GPS-inferred TWS compared with non-constraint observation) assessment are considered. The application of GRACE or GLDAS constraint reduces the bias by 45% and 36% for the internal assessment, whereas the bias is diminished by 32% and 49% for the external assessment."

g. What is the impact of the number and distribution of the constraint points?

Response:

(i) The distribution of constraint points will have impact on the uncertainty distribution. The TWS on the place where constraint points is set to have smaller uncertainty.

(ii) So far, only a few constraint points were added. The increase of constraint points did not necessarily lessen the bias between GPS-TWS and GRACE-TWS or GLDAS-TWS, but model fitness dropped. Besides, adding too much constraint points deviates from our research purpose.

4. Editorial

a. Line 350 – something happened

[what happed?!!]

Response: We do not see point in corresponding places in our word document. "Error! Reference source not found." is due to the cross-reference in Word. Based on your comment, we type it manually that you would not see this format error again.

b. Fig. 3 – only one SD distribution is shown, so it cannot be for different methods and constraint conditions

Response: Our intention is to find out the deviation among all methods and constraint conditions. Therefore, we calculated the SD distribution in order to find out where the large uncertainties are presented in space. The explanation was also given.

Reviewer 3 Report

Dear authors,

I reviewed the paper entitled “Akaike’s Bayesian information criterion for the joint inversion of terrestrial water storage using GPS vertical displacements, GRACE and GLDAS in southwest China”.

The paper concerns about a methodology for estimating TWS using a joint estimation of GPS vertical displacements and other sources such as GRACE and GLDAS to constrain the inversion due to uneven spatial distribution of GPS sites or for improving the inversion procedure. To assess the methodology, the authors use the ABIC technique showing the optimal performance in terms of model fitness and inversion model. The results are compared with other techniques as HVCE and TR discussing the pros and cons of using constrains in terms of model fitness, inversion process stability and self-comparison as well as mutual comparison of GPS-derived TWS with the TWS inferred from the other mentioned sources (GRACE and GLDAS).

The paper is very well written and the structure is correct. The introduction provides enough background to focus the study and the objectives are clearly stated. The methodology is well described and the analysis of the GPS data processing is rigorous including noise analysis.

The results are also well exposed although I will indicate minor points related to them below and the discussion of these results is well reasoned. However, in my opinion it should link and contrast the results with other published works enhancing the advantages of their novel approach for the TWS inversion. There is no reference to any published work in the discussion for that.

Minor points:

Figure 1. I would change the color of triangles from red to black as the figure is not very well readable. Some triangles are intermingled with the background.

Lines 145-146. Remove “to” after (2) and (3).

Lines 178 and 188. Remove the point after ”Figure 2”. Also in line 213 after “Figure 3”

Figure 2 caption. Write “Line” starting with a lowercase letter “l”.

Check the number of Figures, as Figure 2 in page 10 should be Figure 4. Then from this on, revise the number and refer to them in the text accordingly.

Figure 2 in page 10 (or Figure 4). Use bold fonts for letters (a), (b)… as they are not well distinguished. Add a graphical scale in all of them. Also, if a small label with it significance is added in the upper left corner or other place, it could help to better read the figures. For example, “No constrains, HVCE” for (a), “GRACE, HVCE” for (b)… and so on.

Check line 250. There is an error citing the correct figure. I do not understand the first paragraph of section 5.2 well as I do not know which figure you are referring.

Line 360. Replace “GPS-inversed” with “GPS-inverted”.

Line 374. Add “and” before GLDAS.

Line 390. “As shown in Figure 2, the GRACE-inferred TWS is 1.67 cm larger than…”. How can it be seen from this Figure 2?

Table 2. Add units in the header.

Line 411. Figure 4b (or the new number) is cited in the text but nothing it is said in for Figure 4a.

Line 464. “With a mean difference of 1.67 cm…” This value is discussed but it is not exposed in the results. How can it be verified?

Author Response

Please read the PDF version of the response and the revised manuscript, because it can't show figures and equations.

Comments and Suggestions for Authors

Dear authors,

I reviewed the paper entitled “Akaike’s Bayesian information criterion for the joint inversion of terrestrial water storage using GPS vertical displacements, GRACE and GLDAS in southwest China”.

The paper concerns about a methodology for estimating TWS using a joint estimation of GPS vertical displacements and other sources such as GRACE and GLDAS to constrain the inversion due to uneven spatial distribution of GPS sites or for improving the inversion procedure. To assess the methodology, the authors use the ABIC technique showing the optimal performance in terms of model fitness and inversion model. The results are compared with other techniques as HVCE and TR discussing the pros and cons of using constrains in terms of model fitness, inversion process stability and self-comparison as well as mutual comparison of GPS-derived TWS with the TWS inferred from the other mentioned sources (GRACE and GLDAS).

The paper is very well written and the structure is correct. The introduction provides enough background to focus the study and the objectives are clearly stated. The methodology is well described and the analysis of the GPS data processing is rigorous including noise analysis.

The results are also well exposed although I will indicate minor points related to them below and the discussion of these results is well reasoned. However, in my opinion it should link and contrast the results with other published works enhancing the advantages of their novel approach for the TWS inversion. There is no reference to any published work in the discussion for that.

Response: Some previous studies paid additional attention to the application of GPS-inferred TWS, such as more sensitive linkage with drought [18] and GRACE scaling correction [17]. However, from our viewpoints, more researches should be probed into the inversion methods itself since it is far from mature. Therefore, our research focuses more on the inversion method itself from proposing novel inversion methods and strategies and reliability of the existing inversion methods.

Presently, no study has dig into the inversion method performance from the aspect of model fitness and uncertainty distribution analysis. This is rather important to judge the feasibility and correctness of the inversion results. Therefore, our study implemented and assessed their inversion methods, as shown in Section 6.1 and 6.2. Several studies have compared with TWS from other observations, such as GRACE, NOAH NLDAS, NOAH GLDAS. However, few researchers have quantitatively compared the bias between GPS-inferred TWS and TWS from other observations. Fu et al. [17] has adopted RMS reduction to evaluate the difference between GPS-inferred TWS and GRACE-inferred TWS. To compared with his result, we can adopt similar metric -- the ratio of root mean squared difference between GPS-inferred TWS and GRACE-inferred TWS to mean GRACE-inferred TWS. It is implemented and shown in the sixth column of Table 2, the Discrepancy Ratio1 term. It is clear that our difference is smaller than the result reported by Fu et al. [17]. However, lack of quantitative comparison statistics, it is hard to compare with the results from the other researcher besides Fu et al. [17]. Therefore, it is rather important that we proposed evaluate GPS-inferred TWS from model fitness, uncertainty and bias with other observations, which can provide unified metrics for different researchers to compare their results.

Nonetheless, their comparison exhibits similar problems in two aspects that GPS-inferred TWS is relatively larger. As for this part, we supplemented it the manuscript as below:

"... Other researchers reported similar problems that the GPS-inferred TWS is generally larger than other observations [16-18]. Fu et al. [17] discovered that GPS-inferred TWS is larger than the GRACE one, with the root-mean-square reduction of 62.3%, implying the remaining discrepancy of 37.7%. To compare with his results, the ratio of root-mean-square difference between GPS-inferred TWS and GRACE-inferred TWS to the mean of GRACE-inferred TWS, serving as the discrepancy metric, is calculated in the sixth column of Table 2, denoting as Discrepancy Ratio1 term. It is obvious that our discrepancy estimates are smaller than that reported by Fu et al. [17], in particular the case when “pseudo GPS sites” is introduced. Note that GRACE suffers from spatial and temporal aliasing errors [6,7], and leakage effects, while GLDAS merely considers snow water, canopy water, soil water within 2 meters. Therefore, it is hard to judge whether the GPS-inferred TWS is overestimated due to the sensitivity of different observed and modeled data." in line 509 - 520 of the revised manuscript.

6.             Seo, K.-W.; Wilson, C.R.; Chen, J.; Waliser, D.E. GRACE's spatial aliasing error. Geophysical Journal International 2008, 172, 41-48.

7.             Wiese, D.N.; Visser, P.; Nerem, R.S. Estimating low resolution gravity fields at short time intervals to reduce temporal aliasing errors. Advances in Space Research 2011, 48, 1094-1107.

16.           Argus, D.F.; Fu, Y.; Landerer, F.W. Seasonal variation in total water storage in California inferred from GPS observations of vertical motion. Geophysical Research Letters 2014, 41, 1971–1980.

17.           Fu, Y.; Argus, D.F.; Landerer, F.W. GPS As an independent measurement to estimate terrestrial water storage variations in Washington and Oregon. Journal of Geophysical Research Solid Earth 2015, 120, 552-566.

18.           Jin, S.; Zhang, T. Terrestrial Water Storage Anomalies Associated with Drought in Southwestern USA from GPS Observations. Surveys in Geophysics 2016, 37, 1139-1156.

Minor points:

Figure 1. I would change the color of triangles from red to black as the figure is not very well readable. Some triangles are intermingled with the background.

Response: Thank you for your suggestion. We changed the triangles to dark blue in Figure 1.

Lines 145-146. Remove “to” after (2) and (3).

Response: Yes, we did it accordingly.

Lines 178 and 188. Remove the point after ”Figure 2”. Also in line 213 after “Figure 3”

Response: We do not see point in corresponding places in our word document. This is due to the cross-reference in Word. Based on your comment, we type it manually that you won’t see this format error again.

Figure 2 caption. Write “Line” starting with a lowercase letter “l”.

Response: Thank you for your careful checking. We corrected it accordingly.

Check the number of Figures, as Figure 2 in page 10 should be Figure 4. Then from this on, revise the number and refer to them in the text accordingly.

Response: We added a new Figure 3 describing the integral surface extension and grid partitioning. Therefore, all the figure number and relevant citations have been updated.

Figure 2 in page 10 (or Figure 4). Use bold fonts for letters (a), (b)… as they are not well distinguished. Add a graphical scale in all of them. Also, if a small label with it significance is added in the upper left corner or other place, it could help to better read the figures. For example, “No constrains, HVCE” for (a), “GRACE, HVCE” for (b)… and so on.

Response: Given your suggestions, we refined the figures accordingly, as displayed in Figure 5.

Figure 5 The TWS inferred from GRACE, GLDAS, and GPS with different inversion methods and constraint conditions. Figures of the first, second, and third row denote inversion results from HVCE (a, b, c), TR (d, e, f),and ABIC (g, h, i), respectively. The figures in the fourth row are the TWS inferred from GRACE (i) and GLDAS (k). Figures of the first, second and third columns denote inversion results without constraint (a, d, g), with GRACE constraint (b, e, h) and with GLDAS constraint (c, f, i), respectively. The dark-green diamonds denote the location of "pseudo GPS sites".

Check line 250. There is an error citing the correct figure. I do not understand the first paragraph of section 5.2 well as I do not know which figure you are referring.

Response: We think you might mean line 350 --" Error! Reference source not found."

Based on your comment, we type it manually that you won’t see this format error again and explicitly stated the sub-figures.

Line 360. Replace “GPS-inversed” with “GPS-inverted”.

Response: Yes, we did it accordingly.

Line 374. Add “and” before GLDAS.

Response: Yes, we changed accordingly.

Line 390. “As shown in Figure 2, the GRACE-inferred TWS is 1.67 cm larger than…”. How can it be seen from this Figure 2?

Response: You are right. We changed accordingly in line 405-406 as below:

The GRACE-inferred TWS is larger than that of GLDAS in most area, as displayed in Figure 5j and Figure 5k.”

Table 2. Add units in the header.

Response: Thank you for your reminder, we added units in Table 2.

Line 411. Figure 4b (or the new number) is cited in the text but nothing it is said in for Figure 4a.

Response: Thank you for your careful checking, we added description of Figure 7a in the next sentence, which was meant to discuss about TR inversion performance.

Line 464. “With a mean difference of 1.67 cm…” This value is discussed but it is not exposed in the results. How can it be verified?

 Response: You are right. The “mean difference of 41.75px” is our calculation results, which cannot be seen from the Figure 5. However, it is not hard to find out that GLDAS-inferred TWS is smaller than GRACE-inferred TWS by visual inspection from Figure 5. So, we revised it in the manuscript in line 481-483 as below:

“Comparing Figure 5j and Figure 5k, it is not hard to discover that GLDAS-inferred TWS is smaller than that from GRACE, and the former one has more distinct regional characteristics than the smoother GRACE-inferred TWS.”

Round 2

Reviewer 2 Report

I think that the corrections and additions are sufficient

Reviewer 3 Report

Dear authors,

I am satisfied with the changes performed to the paper and the explanations you provided. Congratulations for the paper.